# Optimum Matching Model Using Long-Term Computing on Safer Rural Domestic Water Supply Based on Rainwater Harvesting

**DOI:** 10.3390/ijerph15122864

**Published:** 2018-12-14

**Authors:** Yan-Zhao Jin, Lu-Wen Zhou, Kwong Fai Andrew Lo

**Affiliations:** 1Gansu Research Institute for Water Conservancy, Lanzhou 730000, Gansu, China; jyzgssky@163.com (Y.-Z.J.); zhouluwensky@163.com (L.-W.Z.); 2Graduate Institute of Earth Science, Chinese Culture University, Taipei 11114, Taiwan

**Keywords:** rural domestic water supply, rainwater utilization, long-term method, water balance, optimum matching pattern

## Abstract

A safe rural domestic water supply project has been initiated based on different consumption uses. Long-term computation method and the water balance principle are used to analyze the yearly water demand. Water supply and demand balance is achieved through regulated planning of the rainwater collection surface area and water storage capacity. The best combination of collection area and storage capacity is then determined for various rainfall zones in order to satisfy safe domestic water needs. Ultimately, an optimum matching model is developed to utilize rainwater harvesting for providing safe domestic water in rural areas.

## 1. Introduction

In Northwest China, the annual precipitation ranges between 300 and 550 mm, and total water resources equate to less than 1000 m^3^ per capita. Moreover, the precipitation is unfavorably distributed: 50–70% occurs in July to September, with only 19–24% in May and June, which is the main crop growth period [1]. Owing to the topographic and geological conditions, water conveyance systems are very difficult to build. Agriculture is mainly rain-fed, making it vulnerable and unreliable.

The northern region of China has a long history of drought and water shortage. Historical records show that 749 droughts occurred in the period 206 BC–1949 AD (2155 years), but in the 41 years from 1949–1990, 31 droughts happened, and 7 of these were severe [1].

In recent years, rainwater harvesting systems have been introduced in many countries to solve water supply problems [2,3,4]. Rainwater harvesting techniques have been utilized to solve the water supply problem in rural areas in China [1], Thailand [5], and Jordan [6]. To implement safe domestic water policies in rural areas [7,8], areas with relatively adequate water resources have already developed public water supply systems to combat water shortage difficulties in rural areas. In some remote, scattered areas in the northern region of China where public water systems are not prevalent, rainwater harvesting remains the sole option to satisfy domestic water needs [9,10]. The local rural population have traditionally used rainwater in China. The history of rainwater utilization can be traced back a thousand years. Underground water tanks have been widely built by the local people to store runoff, and rainwater harvesting is widely accepted. The local population has also mastered the techniques of building traditional water cellars, water caves, and upgraded underground water storage tanks.

Owing to the unfavorable conditions, such as low precipitation, uneven rainfall distribution, and low frequency of runoff-producing events, the efficiency of rainwater harvesting in the past was very low, and it could not even meet the demand for domestic use, let alone for supplemental irrigation. The best strategy to enhance the rainwater harvesting efficiency is to concentrate the rainwater both spatially and temporally. The key solution is to construct sufficient rainwater collection surface, water storage, and sediment basins and filtration devices in order to guarantee safe domestic water supply. In fact, the sizing of rainwater storage tanks has been studied in many parts of the world, such as in Italy [11], Brazil [12], Canada [13], Greece [14], and Australia [15]. As such, it is very critical to initiate a safe rural domestic water supply project. So that the dual needs for water quantity and quality may be satisfied by the best combination of collection surface and catchment system capacity, we aim to develop an optimum matching model approach to an economical and safe domestic water supply in the rural areas.

## 2. Materials and Methods

### 2.1. Safe Rural Domestic Water Consumption Uses and Water Use Standards

#### 2.1.1. Consumption Uses

According to the current rural situation in China, the major consumption uses of a safe rural domestic water supply are in the household and in animal husbandry. Nowadays, the household size is becoming smaller with only about 4 to 6 persons per family. On the other hand, large animal husbandry is decreasing to about 1 head per household on average, due to the fact that the farm mechanization level is on the rise.

Small animal husbandry consisting of pig, lamb, and chicken exists only to satisfy the rural population’s meat and egg demands, not for commercial production. Therefore, the total size is rather small. On the average, each household owns 1 pig, 5 sheep, and a total of 10 chickens. This study estimates water consumption on the basis of each such household (Table 1).

#### 2.1.2. Water Use Standard

In 2010, the “Guidelines for Rainwater Catchment Utilization Engineering Technology (GB/T 50596-2010)” [16] was established by many experts appointed by the national government to provide an official directive for rainwater harvesting use in China. The standard for rural domestic water use according to the Guidelines is given in Table 2. The standard for animal husbandry water use is given in Table 3.

The water use standards listed in the Guidelines are usually lower than the actual water use standards commonly practiced by the rural population. However, considering the traditional water use practice and the severe water shortage situation in the northern region of China, the average water use standard according to the Guidelines is used for the subsequent data computation and analyses.

### 2.2. Safe Rural Domestic Water Demand

Rural domestic water use is characterized by an uninterrupted water supply throughout the year in order to achieve the safe rural water supply requirement. The formula to compute the annual safe domestic water supply for each rural household is [17]
(1)$$W_{x}=0.365\overset{n}{\underset{i=1}{\sum }}P_{i}.\;m_{i}$$
where:*W_x_* = annual safe domestic water demand per household (m^3^);*P_i_* = number of actual water consumers (person or head);*m_i_* = standard of actual water consumption (L/d per person or head);*n* = number of consumption uses.

The monthly rural domestic water demands under different water consumption use options are listed in Table 4.

From Table 4, for areas with annual rainfall between 250–500 mm, the estimated annual water demands for rural domestic water consumption use options I, II, and III are about 83.7, 95.2, and 105.9 m^3^, respectively. For areas with annual rainfall exceeding 500 mm, the estimated annual water demands for rural domestic water consumption use options I, II, and III are about 113.1, 131.7, and 149.6 m^3^, respectively.

### 2.3. Rainfall Characteristics of Representative Weather Stations

Two stations—one located at Anding district, Dingxi County with 52 years (1958–2009) of rainfall records, the other at Sifang district, Hsinyang County with 59 years (1951–2009) of rainfall records—were used to optimize rainwater harvesting utilization for the safe rural domestic water supply computation and analysis (Figure 1).

The results indicate that for the two rainfall stations at Anding and Sifang, the mean annual rainfall is about 397.4 mm and 544.0 mm, respectively. The 50% rainfall probabilities for these two stations are 387.8 mm and 521.7 mm, respectively; the 75% rainfall probabilities are about 352.4 mm and 459.6 mm, respectively; and the 90% rainfall probabilities are about 308.1 mm and 425.2 mm, respectively. The long-term monthly rainfall distributions for both stations are listed in Table 5. According to the long-term monthly rainfall distribution (Table 5), the wet season (June to September) comprises 67% and 66.8% of the total annual rainfall at Anding and Sifang, respectively, and the other months (October to May the following year) comprise only about 33% and 33.2% of the total annual rainfall, respectively.

## 3. Results and Discussions

### 3.1. Amount of Rainwater Collection

#### 3.1.1. Current Status of Collection Surfaces

In the northern region of China, the distribution of collection surfaces is currently about 80–120 m^2^ for each household, based on the total area of the tilted farmhouse roof-top. In order to simplify for computational purposes, this study uses a 100 m^2^ tilted roof-top for calculating the collection surface area for each household.

In addition, each rural household usually has an individual courtyard. Usually, it is not waterproofed, so the collection efficiency is very low. Impervious treatment with concrete may add to the domestic supply collection surface area.

#### 3.1.2. Water Supply Reliability Rate

According to the “Guidelines of Rainwater Catchment Utilization Engineering Technology (GB/T 50596-2010)”, the water supply reliability rate is set at 90% in order to satisfy the rural domestic water supply using rainfall harvesting.

#### 3.1.3. Rainwater Collection Efficiency

According to the “Guidelines of Rainwater Catchment Utilization Engineering Technology (GB/T 50596-2010)”, the collection efficiency may differ with different collection materials and rainfall patterns. For areas with 250–500 mm rainfall, the ranges of collection efficiency on a bare-soil courtyard, tilted roof, and concrete surface are about 15–30%, 30–40%, and 73–80%, respectively. For areas with rainfall exceeding 500 mm, the ranges of collection efficiency on a bare-soil courtyard, tilted roof, and concrete surface are about 30–40%, 40–50%, and 75–85%, respectively. Based on related experimental research results as well as empirical extension experience, the rainwater collection efficiencies for different collection materials and rainfall patterns are listed in Table 6.

#### 3.1.4. Assessment of Rainwater Collection

The collection surface is considered as the “water source” of rainwater harvesting for providing a safe rural domestic water supply. This is very critical for ensuring a safe water supply. However, depending on just the tiled roof-top and bare soil courtyard may not adequately satisfy the safe water supply needs. New and high-level collection surfaces must be designed to increase the collection efficiency and the total rainwater collection amount. The following equation can be used to compute the collection amount based on the long-term computation method:(2)$$W_{j}=10^{-3}.P_{p}.\overset{m}{\underset{j}{\sum }}S_{j}.E_{j}$$
where:*W_j_* = rainwater collection amount (m^3^);*P_p_* = rainfall amount (mm);*S_j_* = *j*th type collection surface area (m^2^);*E_j_* = *j*th type collection efficiency (%).

### 3.2. Determination of Matching Facility Design

The optimization procedure consists of two parts. The first part considers the facility’s water supply reliability, and the second part compares the construction cost of the facility.

#### 3.2.1. Optimization of Water Supply Reliability

The preliminary rainwater collection surface area and storage capacity is estimated according to the long-term rainwater harvesting utilization calculation [18] and is based on the above rainwater collection demand for different consumption use options computed using the storage capacity recommended by the Guidelines of Rainwater Catchment Utilization Engineering Technology. The water input, zero initial storage capacity, and the long-term computation method for each assessment time period were used to calculate the potential water supply. The water supply assessment (excess/deficit) for each time period was accumulated at the end of each month to compute the total water storage. When the accumulated storage becomes negative, water shortage occurs during this time period. At the same time, the monthly water supply reliability rate was then determined for the time period. The collection area/storage capacity combination that satisfies the designed water supply reliability rate becomes the first optimum combination.

This computation procedure is then repeated by adjusting the initial rainwater collection surface area and storage capacity up and down in order to determine another optimum combination. As such, the optimum collection area and storage capacity combinations are listed in Table 7 for areas with different rainfall amounts and designed water supply reliability rates.

#### 3.2.2. Optimization of Construction Cost

For a fixed time period, the cost of the operation and maintenance of the rainwater harvesting utilization system remains unchanged. Only the construction material economic cost may change. The unit construction cost of the concrete collection surface is assumed to be at about ¥35/m^2^ (1 ¥ = 0.158 US$). The unit construction cost of a water kiln with a sand/concrete mixed surface is assumed to be about ¥110/m^3^. Therefore, the construction costs of different optimum combinations were computed and are listed in Table 7 also.

The investment in a rainwater harvesting utilization system includes funds provided by the government and the householders’ contributions mainly in the form of labor and local construction materials. According to the statistics [16], out of the total investment, households input 83% toward the domestic supply and 63% toward irrigation supply.

Two aspects may be included in the benefits of the domestic water supply: namely, labor saved in fetching water before the rainwater harvesting utilization system is built and avoidance of water tariff payments. Before building the system, each family has to spend on average 70 labor days fetching water [16]. After installation, this labor could be used to find work in the city and get cash payments. Another benefit is that in the case of a domestic water supply, no water tariff needs to be paid by the households as compared with the pipe water supply. Households can therefore save payments of about ¥2.5/m^3^ (the price of piped water in 2010).

The indirect benefits of the domestic water supply should also include the improvement of health and reduction of costs for medical treatment, which are difficult to express in monetary terms. Besides this, the costs for poverty alleviation and emergency relief incurred by the government during severe droughts also need to be included.

Taking the service life of the rainwater harvesting utilization system to be 20 years and comparing the costs and benefits, the system is highly cost-effective for rural development.

## 4. Conclusions

The study results indicate that in areas with an average of 250–500 mm annual rainfall, the optimum collection area and storage capacity combinations for the water consumption options I, II, and III are (235 m^2^, 40 m^3^), (273 m^2^, 45 m^3^), and (325 m^2^, 45 m^3^), respectively. For areas with average annual rainfall exceeding 500 mm, the optimum combinations are (213 m^2^, 50 m^3^), (267 m^2^, 55 m^3^), and (330 m^2^, 60 m^3^), respectively. The optimum combination model based on the water supply/demand reliability and minimum construction cost produces affordable and cost-effective solutions by reducing the construction investment, saving pipe water tariffs, and increasing cash incomes from extra work opportunities. It also enhances the rainwater harvesting system utilization efficiency for providing a safe rural domestic water supply. Besides this, it may provide needed promotion of rural rainwater harvesting utilization systems and guiding principles for the realization of sustainable development using rainwater harvesting in countries with water shortages.

In general, a wise arrangement of the rainwater harvesting utilization system can guarantee system efficiency and reduced cost. It is essential to design the system according to the local climatic, topographic, geologic, and economic conditions. The capacities of the catchment and storage should be matched to each other to avoid inadequacy in the catchment–storage combination and low system efficiency. Existing impermeable surfaces such as tiled roofs and bare courtyards should be used to store rainwater in storage tanks close to water usage points and to maximize economic advantages.

## Figures and Tables

**Figure 1 ijerph-15-02864-f001:**
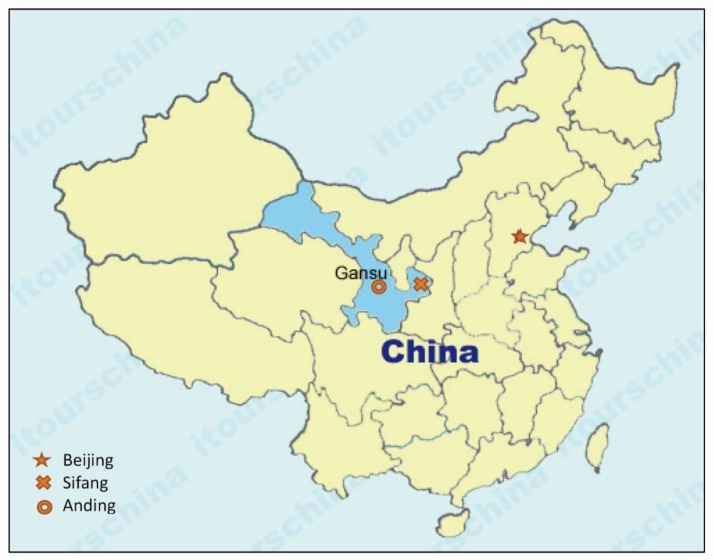
Location map of Gansu Province and the weather stations at Anding and Sifang.

**Table 1 ijerph-15-02864-t001:** Types of water consumption use in the northern region of China.

Consumption Option	Size of Household (Person)	Large Animal (Head)	Small Animal (Head)		
			Pig	Sheep	Chicken
I	4	1	1	5	10
II	5	1	1	5	10
III	6	1	1	5	10

**Table 2 ijerph-15-02864-t002:** Standard for a rainwater harvesting utilization system for household domestic water use.

Rainfall Zone	Water Use Standard (L/d/person)
Average annual rainfall 250–500 mm	20–40
Average annual rainfall >500 mm	40–60

**Table 3 ijerph-15-02864-t003:** Standard for a rainwater harvesting utilization system for household animal husbandry water use.

Type of Animal	Large Animal	Pig	Sheet	Chicken
Water Use Standard (L/d/head)	30–50	20–30	5–10	0.5–1.0

**Table 4 ijerph-15-02864-t004:** Monthly rural household water use (m^3^) based on different consumption use options.

Long-Term Average Annual Rainfall	Water Use Option	Month												Total
		1	2	3	4	5	6	7	8	9	10	11	12	
250–500 mm	I	7.1	6.4	7.1	6.9	7.1	6.9	7.1	7.1	6.9	7.1	6.9	7.1	83.7
	II	8.1	7.3	8.1	7.8	8.1	7.8	8.1	8.1	7.8	8.1	7.8	8.1	95.2
	III	9.0	8.1	9.0	8.7	9.0	8.7	9.0	9.0	8.7	9.0	8.7	9.0	105.9
>500 mm	I	9.6	8.7	9.6	9.3	9.6	9.3	9.6	9.6	9.3	9.6	9.3	9.6	113.1
	II	11.2	10.1	11.2	10.8	11.2	10.8	11.2	11.2	10.8	11.2	10.8	11.2	131.7
	III	12.7	11.5	12.7	12.3	12.7	12.3	12.7	12.7	12.3	12.7	12.3	12.7	149.6

**Table 5 ijerph-15-02864-t005:** Monthly distributions and return probabilities of rainfall at both stations.

Rainfall Zone	Prob. *	Item	Month												Total
			1	2	3	4	5	6	7	8	9	10	11	12	
250–500 mm	Mean	Rainfall (mm)	2.9	4.2	11.3	26.7	46.7	51.6	77.4	83.1	54.3	31.5	6.2	1.5	397.4
		Ratio (%)	0.7	1.1	2.8	6.7	11.8	13.0	19.5	20.9	13.7	7.9	1.6	0.4	100.0
	P = 50%	Rainfall (mm)	2.5	1.0	16.2	20.5	13.8	89.4	62.1	63.9	49.5	66.9	1.3	0.7	387.8
		Ratio (%)	0.6	0.3	4.2	5.3	3.6	23.1	16.0	16.5	12.8	17.3	0.3	0.2	100.0
	P = 75%	Rainfall (mm)	0.0	10.9	7.7	18.6	37.8	28.8	68.0	87.4	82.2	10.7	0.3	0.0	352.4
		Ratio (%)	0.0	3.1	2.2	5.3	10.7	8.2	19.3	24.8	23.3	3.0	0.1	0.0	100.0
	P = 90%	Rainfall (mm)	1.0	2.9	8.3	8.1	24.2	57.9	38.6	38.6	107.3	11.5	9.7	0.0	308.1
		Ratio (%)	0.3	0.9	2.7	2.6	7.9	18.8	12.5	12.5	34.8	3.7	3.1	0.0	100.0
>500 mm	Mean	Rainfall (mm)	4.5	7.5	17.8	35.3	53.1	61.1	115.1	101.6	85.6	42.5	15.9	4.0	544.0
		Ratio (%)	0.8	1.4	3.3	6.5	9.8	11.2	21.2	18.7	15.7	7.8	2.9	0.7	100.0
	P = 50%	Rainfall (mm)	2.9	1.6	30.5	69.8	107.4	18.7	159.4	77.9	23.1	30.4	0.0	0.0	521.7
		Ratio (%)	0.6	0.3	5.8	13.4	20.6	3.6	30.6	14.9	4.4	5.8	0.0	0.0	100.0
	P = 75%	Rainfall (mm)	1.7	12.8	18.1	9.6	48.6	34.0	114.7	125.5	35.5	24.8	32.5	1.8	459.6
		Ratio (%)	0.4	2.8	3.9	2.1	10.6	7.4	25.0	27.3	7.7	5.4	7.1	0.4	100.0
	P = 90%	Rainfall (mm)	1.6	10.2	28.5	18.5	35.7	17.1	110.3	109.4	58.2	21.8	13.9	0.0	425.2
		Ratio (%)	0.4	2.4	6.7	4.4	8.4	4.0	25.9	25.7	13.7	5.1	3.3	0.0	100.0

* Prob. = Return probability.

**Table 6 ijerph-15-02864-t006:** Rainwater collection efficiencies for different surface materials and rainfall zones.

Rainfall Zone	Station	Bare-Soil Courtyard	Tilted Roof-Top (%)	Concrete Surface (%)
250–500 mm	Anding	20	35	77
>500 mm	Sifang	30	45	82

**Table 7 ijerph-15-02864-t007:** Optimum design combinations and construction costs of safe rural domestic water systems.

Rainfall Zone	Water Supply Option	Combination I			Combination II			Combination III			Combination IV			Combination V		
		Area (m^2^)	Storage (m^3^)	Cost (¥)	Area (m^2^)	Storage (m^3^)	Cost (¥)	Area (m^2^)	Storage (m^3^)	Cost (¥)	Area (m^2^)	Storage (m^3^)	Cost (¥)	Area (m^2^)	Storage (m^3^)	Cost (¥)
250–500 mm	I	225	55	13,925	227	55	13,995	228	45	12,930	235 *	40 *	12,625	250	35	12,600,
	II	263	60	15,805	265	55	15,325	267	50	14,845	275 *	45 *	14,575	293	40	14,655
	III	300	60	17,100	303	55	16,655	310	50	16,350	325 *	45 *	16,325	379	40	17,665
>500 mm	I	197	60	13,495	205	55	13,225	213 *	50 *	12,955	231	45	13,035	256	40	13,360
	II	247	65	15,795	255	60	15,525	267 *	55 *	15,395	288	60	16,680	322	45	16,220
	III	294	70	17,990	305	65	17,825	320 *	60 *	17,800	343	55	18,055	385	50	18,975

Note: 1. Area is exclusive of tilted roof-top; 2. * denotes optimum design combination.

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
