# Peer review of "Optimum Matching Model Using Long-Term Computing on Safer Rural Domestic Water Supply Based on Rainwater Harvesting"

_ijerph, 2018, doi:10.3390/ijerph15122864_

Round 1
Reviewer 1 Report
Summary
The study investigates the optimisation of drinking water supply through rainwater harvesting in rural areas of China.
Broad comments
The manuscript adds to the growing literature on the topic on rainwater harvesting especially in rural areas in China.
While the manuscript is easy to understand, please check the English as there are a few minor errors.
For instance, in Line 116,
“This is the very critical for ensuring safe water supply.” It should be:
“This is very critical for ensuring safe water supply.”
Specific Comments
Line 48 – Is this guideline, “Guidelines of Rainwater Catchment Utilization Engineering Technology 48 (GB/T 50596-2010)” a standard guideline in China for water supply design in rural areas? Please give a short description of its uses for the benefit of the international community.
Line 50 – Is the water consumption values in Table 2 related to potable (drinking) water use? Or for standard household use (i.e. laundry, shower etc.?) Please clarify.
Line 108 – “… 30-4% and...” Should this be “…30-40% and…”?
Section 3.1 –
It is a bit confusing as to what constitutes a rainwater collection surface. In Section 3.1.1, only tilted roof-tops are mentioned as being used for calculating collection surface area for each household in the study.
But the authors go into detail out the rainwater collection efficiencies for various other surfaces in Section 3.1.3. Are these materials being included in the calculation as well?
Also, what materials are the roof made of? It appears to have a low collection efficiency.
Section 3.2.1 –
What is the reliability of the storage facility used in providing drinking water? With safe drinking water, the reliability has to be 100% or close to 100%.
What tool was used in determining the optimization? Was a spreadsheet used in determining the optimization values? A short description of the computational method used would be beneficial in understanding the calculations carried out.
Table 7 – The results of the optimization results in large storage values. Is this due to the 100% reliability being required for drinking water? Are these storage dimension practical in actuality?
Additional comments
The reviewer believes that the title of the manuscript is misleading, as it refers to “Drinking Water Supply”, which one would think was for human consumption. However, the optimization calculations within the manuscript take into account the water usage for animal husbandry. It would be cleared if the word “Drinking” was removed from the title.
If drinking water is indeed the focus of the authors, then basic treatment of harvested rainwater should be taken into consideration as well.
The manuscript although short and concise lacks a comprehensive literature review (only 9 references) and the methodology used in carrying out the study is very vague. The literature review and methodology should be further broadened to ensure that the manuscript is clear on its aims and objectives.
There is no real discussion on the implications of the study. For instance, are the derived collection areas, storage areas and associated costs feasible for implementation in rural China? Does the government provide any benefits in implementing such systems? A short discussion will help put the results of the study into context and provide further information to the readers.
Author Response
Dear Reviewer,
Thank you for your comments. They are very valuable and helpful for improving our paper. We have studied all the comments carefully and have made corrections which we hope will meet your kind approval. All the changes have been marked in yellow in the revised manuscript so that they are easily visible to you.
The main responses to your comments are as follows:
I do not feel this paper makes a novel or significant contribution to the area of study/research, i.e. Rainwater Harvesting.
Essentially it is an interesting case study which is based around a technical publication, namely, GB/T 50596-2010. 2010.'Guidelines of Rainwater Catchment Utilization Engineering Technology - Reference 7.
Thank you so much for your comments. We have revised the manuscript according to your valuable suggestions.
The paper could be improved:
The literature review needs to be more extensive, in particular, to include publications from a wider geographical perspective.
Place your study in context with other studies, e.g. Londra P, Theocharis A, Baltas E & Tsihrintzis V. (2015) Optimal sizing of Rainwater Harvesting Tanks for Domestic Use in Greece. Water Resources Management, 29(12), 4357-4377.
Thank you so much for your comments. We have added more relevant references to the original manuscripts:
Abdulla, F.A., Al-Shareef, A.W. 2009. Roof rainwater harvesting systems for household water supply in Jordan. Desalination, 243(1–3): 195–207.
Campisano, A., Modica, C. 2012. Optimal sizing of storage tanks for domestic rainwater harvesting in Sicily. Resources Conservation Recycling, 63: 9–16.
Ghisi, E. 2010. Parameters influencing the sizing of rainwater tanks for use in houses. Water Resources. Management, 24: 2381–2403
Guo, Y., Baetz, B. 2007. Sizing of rainwater storage units for green building applications. J. Hydro. Eng., 12(2):.197–205.
Handia, L., Tembo, J.M., Mwiindwa, C. 2003. Potential of rainwater harvesting in urban Zambia. Phys. Chem. Earth., 28: 893–896.
Londra, P., Theocharis, A., Baltas, E. Tsihrintzis, V. 2015. Optimal sizing of Rainwater Harvesting Tanks for Domestic Use in Greece. Water Resources Management, 29(12): 4357-4377.
Mitchell, V.G. 2007. How important is the selection of computational analysis method to the accuracy of rainwater tank behavior modeling? Hydrol. Process, 21: 2850–2861.
Villareal, E.L., Dixon, A. 2005. Analysis of a rainwater collection system for domestic water supply in Ringdansen, Norrköping, Sweden. Build. Envir., 40: 1174–1184.
Zhou, Y., Shao, W., Zhang, T. 2010. Analysis of a rainwater harvesting system for domestic water supply in Zhoushan, China. J. Zhejiang Univ., 11(5): 342–348.
The optimization methodology - section 3.2 needs expanding and describing in more detail.
Thank you so much for your comments. We have revised the manuscript according to your valuable suggestions by modifying the original descriptive to clarify the optimization method:
The optimization procedure consists of two parts. The first part considers the facility’s water supply reliability, and the second part compares the construction cost of the facility.
3.2.1 Optimization of Water Supply Reliability
3.2.2 Optimization of Construction Cost
Responses to Reviewer 1 Comments:
Dear Reviewer,
Thank you for your comments. They are very valuable and helpful for improving our paper. We have studied all the comments carefully and have made corrections which we hope will meet your kind approval. All the changes have been marked in yellow in the revised manuscript so that they are easily visible to you.
The main responses to your comments are as follows:
Summary
The study investigates the optimisation of drinking water supply through rainwater harvesting in rural areas of China.
Broad comments
The manuscript adds to the growing literature on the topic on rainwater harvesting especially in rural areas in China.
Thank you so much for your comments. We have revised the manuscript according to your valuable suggestions.
While the manuscript is easy to understand, please check the English as there are a few minor errors.
For instance, in Line 116,
“This is the very critical for ensuring safe water supply.” It should be:
“This is very critical for ensuring safe water supply.”
Thank you so much for your comments. We have revised the manuscript according to your valuable suggestions.
Specific Comments
Line 48 – Is this guideline, “Guidelines of Rainwater Catchment Utilization Engineering Technology 48 (GB/T 50596-2010)” a standard guideline in China for water supply design in rural areas? Please give a short description of its uses for the benefit of the international community.
Thank you so much for your comments. We have revised the manuscript according to your valuable suggestions as follows:
In 2010, the “Guidelines for Rainwater Catchment Utilization Engineering Technology (GB/T 50596-2010)” was established by many experts appointed by the national government to provide an official directive for rainwater harvesting use in China. According to the Guidelines, the standard for rural domestic water use is listed in Table 2.
Line 50 – Is the water consumption values in Table 2 related to potable (drinking) water use? Or for standard household use (i.e. laundry, shower etc.?) Please clarify.
Thank you so much for your comments. The water consumption is for both potable and household use. To avoid confusion, the term “domestic” water use is used instead of “drinking” or “household” water use throughout the entire text.
Line 108 – “… 30-4% and...” Should this be “…30-40% and…”?
Thank you for your comment. The mistyped value “40%” has been corrected.
Section 3.1 –
It is a bit confusing as to what constitutes a rainwater collection surface. In Section 3.1.1, only tilted roof-tops are mentioned as being used for calculating collection surface area for each household in the study.
But the authors go into detail out the rainwater collection efficiencies for various other surfaces in Section 3.1.3. Are these materials being included in the calculation as well?
Thank you so much for your comments. Both the roof-tops, bare and concrete courtyard surface are used to collect rainwater. As such, the descriptive in Section 3.1.1 is rewritten as:
In the northern region of China, the distribution of collection surface currently is about 80–120 m2 for each household, based on the total area of the tilted farm house roof-top. In order to simplify computational purpose, this study uses 100 m2 tilted roof-top for calculating collection surface area for each household.
In addition, each rural household usually has an individual courtyard. Usually, it is not being waterproof treated. Collection efficiency is very low. Impervious treatment with concrete may add to the domestic supply collection surface.
Also, what materials are the roof made of? It appears to have a low collection efficiency.
Thank you for your comment. The roof materials are usually tilted material. However, due to most of the roof-tops are old-aged and lack of good maintenance, the collection efficiency on the average is not that high.
Section 3.2.1 –
What is the reliability of the storage facility used in providing drinking water? With safe drinking water, the reliability has to be 100% or close to 100%.
Thank you for your comment. The water supply reliability is et at 90%. Originally, this statement is included in Section 3.1.2.
What tool was used in determining the optimization? Was a spreadsheet used in determining the optimization values? A short description of the computational method used would be beneficial in understanding the calculations carried out.
Thank you so much for your comments. To clarify the optimization method, an explanatory paragraph is added right after the Section 3.2 heading and both headings for Sections 3.2.1 and 3.2.2 have been rephrased:
3.2. Determination of Matching Facility Design
The optimization procedure consists of two parts. The first part considers the facility’s water supply reliability, and the second part compares the construction cost of the facility.
3.2.1 Optimization of Water Supply Reliability
3.2.2 Optimization of Construction Cost
Table 7 – The results of the optimization results in large storage values. Is this due to the 100% reliability being required for drinking water? Are these storage dimension practical in actuality?
Thank you so much for your comments. Yes, the high storage values are due to high reliability rate of 90%. However, these dimension is very practical in rural areas in China.
Additional comments
The reviewer believes that the title of the manuscript is misleading, as it refers to “Drinking Water Supply”, which one would think was for human consumption. However, the optimization calculations within the manuscript take into account the water usage for animal husbandry. It would be cleared if the word “Drinking” was removed from the title.
If drinking water is indeed the focus of the authors, then basic treatment of harvested rainwater should be taken into consideration as well.
Thank you so much for your comments. Accordingly, the word “Drinking” in the paper title has been replaced by “Domestic”.
The manuscript although short and concise lacks a comprehensive literature review (only 9 references) and the methodology used in carrying out the study is very vague. The literature review and methodology should be further broadened to ensure that the manuscript is clear on its aims and objectives.
Thank you so much for your comments. Accordingly, additional references have been included in the introduction section:
In recent years, rainwater harvesting systems have been introduced in many countries to solve the water supply problems [1,2,3]. Rainwater harvesting technique has been utilized to solve the water supply problem in rural areas in China, Thailand and Jordan [4,5,6]. The northern region of China has a long history of drought and water shortage. To implement the safe domestic water policy in rural area [7,8], areas with relatively adequate water resources have already developed public water supply systems to combat water shortage difficulties in rural areas. In some remote, scattered areas in the northern region of china where public water systems are not prevalent, rainwater harvesting remains as the sole option to satisfy the domestic water needs [9,10]. The key solution is to construct enough rainwater collection surface, water storage, along with sediment basins and filtration devices in order to guarantee safe domestic water supply. In fact, sizing of rainwater storage tanks has been studied in many parts of the world [11,12,13,14,15]. As such, it is very critical to initiate a safe rural domestic water supply project. So that the dual needs for water quantity and quality safety may be satisfied by the best combination of collection surface and catchment system capacity, in hope of developing an economical and safe domestic water based on optimum matching model approach in the rural areas.
Added references:
Abdulla, F.A., Al-Shareef, A.W. 2009. Roof rainwater harvesting systems for household water supply in Jordan. Desalination, 243(1–3): 195–207.
Campisano, A., Modica, C. 2012. Optimal sizing of storage tanks for domestic rainwater harvesting in Sicily. Resources Conservation Recycling, 63: 9–16.
Ghisi, E. 2010. Parameters influencing the sizing of rainwater tanks for use in houses. Water Resources. Management, 24: 2381–2403
Guo, Y., Baetz, B. 2007. Sizing of rainwater storage units for green building applications. J. Hydro. Eng., 12(2):.197–205.
Handia, L., Tembo, J.M., Mwiindwa, C. 2003. Potential of rainwater harvesting in urban Zambia. Phys. Chem. Earth., 28: 893–896.
Londra, P., Theocharis, A., Baltas, E. Tsihrintzis, V. 2015. Optimal sizing of Rainwater Harvesting Tanks for Domestic Use in Greece. Water Resources Management, 29(12): 4357-4377.
Mitchell, V.G. 2007. How important is the selection of computational analysis method to the accuracy of rainwater tank behavior modeling? Hydrol. Process, 21: 2850–2861.
Villareal, E.L., Dixon, A. 2005. Analysis of a rainwater collection system for domestic water supply in Ringdansen, Norrköping, Sweden. Build. Envir., 40: 1174–1184.
Zhou, Y., Shao, W., Zhang, T. 2010. Analysis of a rainwater harvesting system for domestic water supply in Zhoushan, China. J. Zhejiang Univ., 11(5): 342–348.
There is no real discussion on the implications of the study. For instance, are the derived collection areas, storage areas and associated costs feasible for implementation in rural China? Does the government provide any benefits in implementing such systems? A short discussion will help put the results of the study into context and provide further information to the readers.
Thank you so much for your comments. An implication descriptive is added towards the end of the Results and Discussions Section:
This optimization model based on the water supply demand reliability and minimum construction cost, besides being cost effective, may enhance rainwater utilization efficiency, encourage rainfall harvesting of rural domestic water supply, and sustainable development of rainwater resources in the future.
Reviewer 2 Report
I do not feel this paper makes a novel or significant contribution to the area of study/research, i.e. Rainwater Harvesting.
Essentially it is an interesting case study which is based around a technical publication, namely, GB/T 50596-2010. 2010.'Guidelines of Rainwater Catchment Utilization Engineering Technology - Reference 7.
The paper could be improved:
The literature review needs to be more extensive, in particular, to include publications from a wider geographical perspective.
Place your study in context with other studies, e.g. Londra P, Theocharis A, Baltas E & Tsihrintzis V. (2015) Optimal sizing of Rainwater Harvesting Tanks for Domestic Use in Greece. Water Resources Management, 29(12), 4357-4377.
The optimization methodology - section 3.2 needs expanding and describing in more detail.
Author Response
Dear Reviewer,
Thank you for your comments. They are very valuable and helpful for improving our paper. We have studied all the comments carefully and have made corrections which we hope will meet your kind approval. All the changes have been marked in yellow in the revised manuscript so that they are easily visible to you.
The main responses to your comments are as follows:
I do not feel this paper makes a novel or significant contribution to the area of study/research, i.e. Rainwater Harvesting.
Essentially it is an interesting case study which is based around a technical publication, namely, GB/T 50596-2010. 2010.'Guidelines of Rainwater Catchment Utilization Engineering Technology - Reference 7.
Thank you so much for your comments. We have revised the manuscript according to your valuable suggestions.
The paper could be improved:
The literature review needs to be more extensive, in particular, to include publications from a wider geographical perspective.
Place your study in context with other studies, e.g. Londra P, Theocharis A, Baltas E & Tsihrintzis V. (2015) Optimal sizing of Rainwater Harvesting Tanks for Domestic Use in Greece. Water Resources Management, 29(12), 4357-4377.
Thank you so much for your comments. We have added more relevant references to the original manuscripts:
Abdulla, F.A., Al-Shareef, A.W. 2009. Roof rainwater harvesting systems for household water supply in Jordan. Desalination, 243(1–3): 195–207.
Campisano, A., Modica, C. 2012. Optimal sizing of storage tanks for domestic rainwater harvesting in Sicily. Resources Conservation Recycling, 63: 9–16.
Ghisi, E. 2010. Parameters influencing the sizing of rainwater tanks for use in houses. Water Resources. Management, 24: 2381–2403
Guo, Y., Baetz, B. 2007. Sizing of rainwater storage units for green building applications. J. Hydro. Eng., 12(2):.197–205.
Handia, L., Tembo, J.M., Mwiindwa, C. 2003. Potential of rainwater harvesting in urban Zambia. Phys. Chem. Earth., 28: 893–896.
Londra, P., Theocharis, A., Baltas, E. Tsihrintzis, V. 2015. Optimal sizing of Rainwater Harvesting Tanks for Domestic Use in Greece. Water Resources Management, 29(12): 4357-4377.
Mitchell, V.G. 2007. How important is the selection of computational analysis method to the accuracy of rainwater tank behavior modeling? Hydrol. Process, 21: 2850–2861.
Villareal, E.L., Dixon, A. 2005. Analysis of a rainwater collection system for domestic water supply in Ringdansen, Norrköping, Sweden. Build. Envir., 40: 1174–1184.
Zhou, Y., Shao, W., Zhang, T. 2010. Analysis of a rainwater harvesting system for domestic water supply in Zhoushan, China. J. Zhejiang Univ., 11(5): 342–348.
The optimization methodology - section 3.2 needs expanding and describing in more detail.
Thank you so much for your comments. We have revised the manuscript according to your valuable suggestions by modifying the original descriptive to clarify the optimization method:
The optimization procedure consists of two parts. The first part considers the facility’s water supply reliability, and the second part compares the construction cost of the facility.
3.2.1 Optimization of Water Supply Reliability
3.2.2 Optimization of Construction Cost
Round 2
Reviewer 1 Report
The manuscript has improved slightly. While the references list has been expanded, the relevance of the references to the manuscript has not been conveyed with much detail.
Furthermore the manuscript only presents the results of the optimisation without discussing in more detail of its implications. The following has been added:
"This optimization model based on the water supply demand reliability and minimum construction cost, besides cost effective, may enhance rainwater utilization efficiency, encourage rainfall harvesting of rural domestic water supply, and sustainable development of rainwater resources in the future."
But it does not provide the readers with any significant information on rainwater harvesting in rural China. The authors may potentially discuss, the storage-cost combination with respect to the rainfall zones, whether poorer farmers can afford the construction costs; suggest government incentives, etc; to name a few. A more succulent discussion is required to provide completeness to the manuscript.
Line 168 – “…(235, 40), (273,45), and (325,45)…” It is better to write it as (235m2, 40m3), (273m2,45m3), and (325m2,45m3). Same for the information in the next line.
Again, the conclusion is too short and does not provide sufficient information to conclude off the manuscript.
Author Response
Dear Reviewer,
Thank you for your comments. They are very valuable and helpful for improving our paper. We have studied all the comments carefully and have made corrections which we hope will meet your kind approval. All the changes have been marked in yellow in the revised manuscript so that they are easily visible to you.
The main responses to your comments are as follows:
1. The manuscript has improved slightly. While the references list has been expanded, the relevance of the references to the manuscript has not been conveyed with much detail.
Thank you so much for your comments. We have revised the introduction section of the manuscript according to your valuable suggestions.
In the Northwest China, the annual precipitation ranges between 300 and 550 mm, and total water resources equate to less than 1,000 m3 per capita. Moreover, the precipitation is unfavorably distributed: 50 -70% occurring in July to September, with only 19-24% in May and June that is the main crop growth period [1]. Owing to the topographic and geological conditions, water conveyance systems are very difficult to build. Agriculture is mainly rain-fed making it vulnerable and unreliable.
The northern region of China has a long history of drought and water shortage. Historical records show that 749 droughts occurred in the period 206 BC – 1949 AD (2,155 years), but in the 41 years from 1949 – 1990, there were 31 droughts happened, and seven of these were severe [1].
In recent years, rainwater harvesting systems have been introduced in many countries to solve the water supply problems [2, 3, 4]. Rainwater harvesting technique has been utilized to solve the water supply problem in rural areas in China [1], Thailand [5] and Jordan [6]. To implement the safe domestic water policy in rural area [7, 8], areas with relatively adequate water resources have already developed public water supply systems to combat water shortage difficulties in rural areas. In some remote, scattered areas in the northern region of china where public water systems are not prevalent, rainwater harvesting remains as the sole option to satisfy the domestic water needs [9, 10]. The local rural population have traditionally used rainwater in China. The history of rainwater utilization can be traced back a thousand years. Underground water tanks have been widely built by the local people to store runoff, and rainwater harvesting is widely accepted. The local population has also mastered the techniques of building the traditional water cellar, water cave, and upgraded underground water storage tank.
Owing to the unfavorable conditions, such as low precipitation, uneven rainfall distribution, and low runoff producing events, the efficiency of rainwater harvesting in the past was very low, and it could not even meet the demand for domestic use, let alone for supplemental irrigation. The best strategy to enhance the rainwater harvesting efficiency is to concentrate the rainwater both spatially and temporally. The key solution is to construct enough rainwater collection surface, water storage, along with sediment basins and filtration devices in order to guarantee safe domestic water supply. In fact, sizing of rainwater storage tanks has been studied in many parts of the world, such as in Italy [11], in Brazil [12], in Canada [13], in Greece [14], and in Australia [15]. As such, it is very critical to initiate a safe rural domestic water supply project. So that the dual needs for water quantity and quality safety may be satisfied by the best combination of collection surface and catchment system capacity, in hope of developing an economical and safe domestic water based on optimum matching model approach in the rural areas.
2. Furthermore the manuscript only presents the results of the optimisation without discussing in more detail of its implications. The following has been added:
"This optimization model based on the water supply demand reliability and minimum construction cost, besides cost effective, may enhance rainwater utilization efficiency, encourage rainfall harvesting of rural domestic water supply, and sustainable development of rainwater resources in the future."
But it does not provide the readers with any significant information on rainwater harvesting in rural China. The authors may potentially discuss, the storage-cost combination with respect to the rainfall zones, whether poorer farmers can afford the construction costs; suggest government incentives, etc; to name a few. A more succulent discussion is required to provide completeness to the manuscript.
Thank you so much for your comments. We have revised the discussion section of the manuscript according to your valuable suggestions.
The investment of the rainwater harvesting utilization system included fund provided by the government and the householders’ contributions mainly in the form of labor and local construction materials. According to the statistics [16], out of the total investment, households input 83% toward the domestic supply and 63% toward irrigation supply.
Two aspects may be included in the benefits of the domestic water supply: namely, labor saved for fetching water before the rainwater harvesting utilization system is built and avoidance of water tariff payments. Before building the system, each family has to spend on an average 70 labor days for fetching water [16]. After installation, this labor could be used to find work in the city and get cash payment. Another benefit is that in the case of domestic water supply, no water tariff needed to be paid by the households as compared with the pipe water supply. Households can therefore save payment of about ¥ 2.5/m3 (the price of piped water in 2010).
The indirect benefits of domestic water supply should also include improvement of health and reduction of cost for medical treatment, which are difficult to express in monetary terms. Besides, the costs for poverty alleviation and emergency relief incurred by the government during severe droughts also need to be included.
Taking the service life of the rainwater harvesting utilization system as 20 years and comparing the costs and benefits, the system is highly cost-effective for rural development.
3. Line 168 – “…(235, 40), (273,45), and (325,45)…” It is better to write it as (235m2, 40m3), (273m2,45m3), and (325m2,45m3). Same for the information in the next line.
Thank you so much for your comments. We have revised these lines in the manuscript according to your valuable suggestions.
Study results indicate that with average 250–500 mm annual rainfall areas, the optimum collection area and storage capacity combination for the I, II, and III water consumption option is (235 m2, 40 m3), (273 m2,45 m3), and (325 m2,45 m3), respectively. For areas with average annual rainfall exceeding 500 mm, the optimum combination is (213 m2, 50 m3), (267 m2, 55 m3), and (330 m2, 60 m3), respectively.
4. Again, the conclusion is too short and does not provide sufficient information to conclude off the manuscript.
Thank you so much for your comments. We have revised the conclusion section in the manuscript according to your valuable suggestions.
Study results indicate that with average 250–500 mm annual rainfall areas, the optimum collection area and storage capacity combination for the I, II, and III water consumption option is (235 m2, 40 m3), (273 m2,45 m3), and (325 m2,45 m3), respectively. For areas with average annual rainfall exceeding 500 mm, the optimum combination is (213 m2, 50 m3), (267 m2, 55 m3), and (330 m2, 60 m3), respectively. The optimum combination model based on the water supply demand reliability and minimum construction cost is affordable, cost effective by reducing the construction investment, saving pipe water tariff, and increasing extra work cash incomes. It also enhance rainwater harvesting system utilization efficiency for providing safe rural domestic water supply. Besides, it may provide needed promotion of rural rainwater harvesting utilization systems, and guiding principles for realization of sustainable development using rainwater harvesting in water shortage countries.
In general, a wise arrangement of the rainwater harvesting utilization system can guarantee system efficiency and reduced cost. It is essential to design the system according to the local climatic, topographic, geologic and economic conditions. The capacity of catchment and storage should be matched to each other, to avoid inadequacy in catchment-storage combination and low system efficiency. Existing impermeable surfaces such as tiled roof and bare courtyard, should be used to store rainwater in storage tanks closed to water usage points and to maximum the economic advantages.
Reviewer 2 Report
The paper has been improved and includes additional references. However, I still feel the paper reads very much as a case study and does not add significantly to the area.
Author Response
Dear Reviewer,
Thank you for your comments. They are very valuable and helpful for improving our paper. We have studied all the comments carefully and have made corrections which we hope will meet your kind approval. All the changes have been marked in yellow in the revised manuscript so that they are easily visible to you.
The main responses to your comments are as follows:
1. The paper has been improved and includes additional references. However, I still feel the paper reads very much as a case study and does not add significantly to the area.
Thank you so much for your comments. We have revised the introduction, discussions and conclusion section in the manuscript according to your valuable suggestions.
Introduction
In the Northwest China, the annual precipitation ranges between 300 and 550 mm, and total water resources equate to less than 1,000 m3 per capita. Moreover, the precipitation is unfavorably distributed: 50 -70% occurring in July to September, with only 19-24% in May and June that is the main crop growth period [1]. Owing to the topographic and geological conditions, water conveyance systems are very difficult to build. Agriculture is mainly rain-fed making it vulnerable and unreliable.
The northern region of China has a long history of drought and water shortage. Historical records show that 749 droughts occurred in the period 206 BC – 1949 AD (2,155 years), but in the 41 years from 1949 – 1990, there were 31 droughts happened, and seven of these were severe [1].
In recent years, rainwater harvesting systems have been introduced in many countries to solve the water supply problems [2, 3, 4]. Rainwater harvesting technique has been utilized to solve the water supply problem in rural areas in China [1], Thailand [5] and Jordan [6]. To implement the safe domestic water policy in rural area [7, 8], areas with relatively adequate water resources have already developed public water supply systems to combat water shortage difficulties in rural areas. In some remote, scattered areas in the northern region of china where public water systems are not prevalent, rainwater harvesting remains as the sole option to satisfy the domestic water needs [9, 10]. The local rural population have traditionally used rainwater in China. The history of rainwater utilization can be traced back a thousand years. Underground water tanks have been widely built by the local people to store runoff, and rainwater harvesting is widely accepted. The local population has also mastered the techniques of building the traditional water cellar, water cave, and upgraded underground water storage tank.
Owing to the unfavorable conditions, such as low precipitation, uneven rainfall distribution, and low runoff producing events, the efficiency of rainwater harvesting in the past was very low, and it could not even meet the demand for domestic use, let alone for supplemental irrigation. The best strategy to enhance the rainwater harvesting efficiency is to concentrate the rainwater both spatially and temporally. The key solution is to construct enough rainwater collection surface, water storage, along with sediment basins and filtration devices in order to guarantee safe domestic water supply. In fact, sizing of rainwater storage tanks has been studied in many parts of the world, such as in Italy [11], in Brazil [12], in Canada [13], in Greece [14], and in Australia [15]. As such, it is very critical to initiate a safe rural domestic water supply project. So that the dual needs for water quantity and quality safety may be satisfied by the best combination of collection surface and catchment system capacity, in hope of developing an economical and safe domestic water based on optimum matching model approach in the rural areas.
Discussion
The investment of the rainwater harvesting utilization system included fund provided by the government and the householders’ contributions mainly in the form of labor and local construction materials. According to the statistics [16], out of the total investment, households input 83% toward the domestic supply and 63% toward irrigation supply.
Two aspects may be included in the benefits of the domestic water supply: namely, labor saved for fetching water before the rainwater harvesting utilization system is built and avoidance of water tariff payments. Before building the system, each family has to spend on an average 70 labor days for fetching water [16]. After installation, this labor could be used to find work in the city and get cash payment. Another benefit is that in the case of domestic water supply, no water tariff needed to be paid by the households as compared with the pipe water supply. Households can therefore save payment of about ¥ 2.5/m3 (the price of piped water in 2010).
The indirect benefits of domestic water supply should also include improvement of health and reduction of cost for medical treatment, which are difficult to express in monetary terms. Besides, the costs for poverty alleviation and emergency relief incurred by the government during severe droughts also need to be included.
Taking the service life of the rainwater harvesting utilization system as 20 years and comparing the costs and benefits, the system is highly cost-effective for rural development.
Conclusion
Study results indicate that with average 250–500 mm annual rainfall areas, the optimum collection area and storage capacity combination for the I, II, and III water consumption option is (235 m2, 40 m3), (273 m2,45 m3), and (325 m2,45 m3), respectively. For areas with average annual rainfall exceeding 500 mm, the optimum combination is (213 m2, 50 m3), (267 m2, 55 m3), and (330 m2, 60 m3), respectively. The optimum combination model based on the water supply demand reliability and minimum construction cost is affordable, cost effective by reducing the construction investment, saving pipe water tariff, and increasing extra work cash incomes. It also enhance rainwater harvesting system utilization efficiency for providing safe rural domestic water supply. Besides, it may provide needed promotion of rural rainwater harvesting utilization systems, and guiding principles for realization of sustainable development using rainwater harvesting in water shortage countries.
In general, a wise arrangement of the rainwater harvesting utilization system can guarantee system efficiency and reduced cost. It is essential to design the system according to the local climatic, topographic, geologic and economic conditions. The capacity of catchment and storage should be matched to each other, to avoid inadequacy in catchment-storage combination and low system efficiency. Existing impermeable surfaces such as tiled roof and bare courtyard, should be used to store rainwater in storage tanks closed to water usage points and to maximum the economic advantages.
Round 3
Reviewer 1 Report
The paper is much improved although some minor English issues need addressing.
Reviewer 2 Report
The paper is much improved but still requires an edit to address minor English issues. However,
I am still of he opinion this paper would be better as a case study/technical note rather than a full journal article.